# Woaded Blue: A Colorful Approach to the Dialectic between Written Historical Sources, Experimental Archaeology, Chromatographic Analyses, and Biochemical Research

**Dominique Cardon** [1],*, **Zvi C. Koren** [2] **and Hisako Sumi** [3]

1   CIHAM/UMR 5648 CNRS, 14 av. Berthelot, CEDEX 07, 69363 Lyon, France
2   The Edelstein Center for the Analysis of Ancient Artifacts, Department of Chemical Engineering, Shenkar College of Engineering, Design and Art, 12 Anna Frank St., Ramat-Gan 52526, Israel
3   North-Indigo Textile Arts Studio, 2-3-9, Matsugae, Otaru 047-0022, Japan
*   Correspondence: cardon.dominique@wanadoo.fr

**Abstract:** Research into the sustainability of natural, potentially renewable, resources is one of the major issues of our time. It naturally includes the quest for sustainable sources of colorants for textiles, cosmetics, and food. In industrialized countries, natural dyeing with plants and a few species of coccid insects was practiced on a large scale for centuries before synthetic colorants were developed. Therefore, historical documents on the growing of dye plants and dyeing processes offer a relevant basis from which to start reconsidering the potential of natural colorants in our time. However, written sources need to be completed by experimental archaeologists to allow a scientific understanding of the biochemical reactions at work in the historical processes described. The results of such interdisciplinary research can then inspire contemporary programs to revive the production of natural dyes. The long history of dyeing blue with woad, *Isatis tinctoria* L., is revisited here as an illustration of the fruitful complementarity of sources and approaches. This article presents a step-by-step reassessment of the production chain of woad as described in historical texts, from the growing of the plant to its use as a source of indigo in the woad and indigo vats. The experimental reconstitution of the processing of woad leaves into couched woad allowed us to follow the evolution of the composition and proportions of indigoid colorants in the leaves by HPLC analyses. Additionally, HPLC analyses allowed a comparison of the respective indigoid contents of couched woad and *sukumo*, the form of indigo dye resulting from another couching process, traditionally used in Japan for dyers' knotweed, *Persicaria tinctoria* (Ait.) H. Gross. The reconstitution of the 18th century woad and indigo vat process allowed investigations into the bacterial flora associated with the use of couched woad in vat liquors, which were found to contain different indigo-reducing bacteria, including two distinct strains of a new indigo-reducing species.

**Keywords:** natural dyes; indigo; woad; *Isatis tinctoria*; woad balls; couched woad; woad and indigo vat; HPLC; indigoid colorants; indigo-reducing bacteria; Etienne Ferrières's Register; Antoine Janot; Paul Gout; 18th century memoirs on dyeing; reconstitution of dyeing processes

## 1. Introduction—Context and Purpose

In the present global state of the world, it should appear evident that, with the accelerated depletion of fossil energies and resources, the survival of humankind requires better knowledge and hence, intensified research into the sustainability of natural, potentially renewable, resources. The production and uses of natural colorants are a part of this major issue of our time, particularly as obvious sources of colorants for textiles, cosmetics, and food. In industrialized countries, natural dyeing was practiced on a large scale for centuries before the first synthetic colorants were developed. Therefore, historical documents on the growing of dye plants and dyeing offer a relevant basis from which to start reconsidering the potential of natural colorants in our time.

In Europe, historical archives, both public and private, abound in documents on the cultivation of dye plants and dyeing processes. However, these seldom give all the necessary information and even when precise quantitative data are provided, they need to be converted into modern metrological systems and put into proper economic and technological contexts to become meaningful. Further, it is often not obvious to fully grasp the scale of the technical processes described and assess their relevance to present conditions. In such a situation, experimental archaeology provides a most useful complementary approach to check the accuracy and usefulness of historical sources, and to better understand the biochemical reactions at work in the described processes. In addition, experimental archaeology can reveal gaps in historical technical descriptions and inspire contemporary programs to revive the production of natural dyes. To that effect, woad was grown, collected, and processed into couched woad in France by the first author and then sent to Japan to the third author who used it to set a biological woad and indigo vat according to 18th century French dyers' recipes, as part of her experimental studies of a range of biological indigo vats.

This paper presents a case study illustrating the complementarity of these sources and approaches concerning the production of blues via vat dyeing. The focus of this paper is on a particularly important part of the history of the production of blue dyes, by presenting a step-by-step re-assessment of the production of woad and its remarkably long-lasting contribution to indigo vats for wool in Europe.

## 2. Woad for Blue for Wool

Woad, *Isatis tinctoria* L., is well known as the only indigo-producing plant indigenous to Europe and the Middle East. It is, therefore, the probable source of the indigo detected in the fast-growing number of analyzed prehistoric textiles from the Bronze Age and Iron Age, discovered in archaeological sites scattered from Ukraine to Scandinavia [1] (pp. 70–71). These findings reveal the prehistoric antiquity of the "gesamteuropäischen Kultur der Waidherstellung" (meaning "pan-European culture of woad production") already highlighted by the German historian Stephen Selzer for the Medieval Period [2] (p. 341).

Fragments of information scattered in written historical documents of different types, from business accounts to guild regulations from 14th to 16th c. France and Italy, have allowed to reconstitute the whole production process and, at the same time, to collect a growing—albeit heteroclite—corpus of quantitative data [3] (pp. 52–61), [4] (pp. 210–305).

Current knowledge can be summarized as follows. Woad seeds sewn in the spring would give four to five successive crops of fresh leaves, from early summer to the middle of autumn. In the south of France and Italy, a fifth harvest could be collected in autumn but it was usually considered of lesser quality as a source for a blue colorant. Leaves were crushed in horse mills, as soon as possible after having been picked, and the resulting mash was shaped into fist-sized balls, dried in sheds, and stored for local consumption or collected by woad merchants and shipped to other regions and foreign countries.

The form in which woad was finally used as a source of indigo in the blue vats was called couched woad in English and *pastel agranat* in French. It was prepared by specialized artisans, either in the regions of important woad cultivation or in the textile centers of wool broadcloth production, where it was used in large quantities. It was obtained by crushing the dry woad balls on paved ground in special buildings [5] (p. 327). Water was added in successive stages during the following days, and the crumbling mass of woad started fermenting and had to be regularly turned over, until after about three weeks. The mass had turned into small dry lumps (hence the expression *agranat*, meaning agglomerated into grains, in Occitan language). The word "couch" dates in English to about AD 1300, and is derived from the French verb *coucher* which means "to lay down, to spread or lay on a surface, to overlay," from Old French *couchier*, which also has a similar meaning to "to lay down, place" as in "go to bed, put to bed". The crushed woad was spread in a thick bed to allow an even distribution of the water and even fermentation—hence, "couch-

ing". Interestingly, they have the same imagery in Japanese concerning *sukumo*, meaning "couching". The dry leaves are laid in an even mass (like a straw mattress) to make *sukumo* by watering and letting it ferment. They call that "to put the *ai*, indigo, to bed" and the place they make it is called "the indigo's bedroom".

Couched woad was the only source of indigo, i.e., of a fast blue colorant, available and used on a large scale in Europe in the wool broadcloth industry and tapestry weaving until the first part of the 17th century. Even after the 17th c., when the indigo pigment from India or the West Indies began being imported and used in increasingly bigger quantities in the European blue vats for wool, woad went on being the major ingredient in the setting of the vats until the end of the 19th century [6] (p. 42).

## 3. Re-Examining Key Information from the Account Book of a Woad Merchant in Light of Experimental Archaeology

### 3.1. Etienne Ferrières's Registre

The register of Etienne Ferrières (Figure 1), preserved in the Municipal Archives of Toulouse under the reference HH 61, describes Ferrières's business transactions for the years AD 1559 to 1561.

Etienne Ferrières lived in Toulouse, where he was one of the very rich and powerful merchants dealing in woad from Languedoc (the south of France) in the 16th century, at the apex of the international importance of this commodity [7]. In this account book, at folio 25 verso, he provides an important piece of quantitative information that was nowhere else to be found until recently: the weight of couched woad obtained from the processing of a number of woad balls.

The circumstances in which he writes are exceptional; in Languedoc, the summer of 1559 has been the driest in living memory. Consequently, the woad crop is going to be much reduced in quantities, in comparison with normal years, but the quality will be excellent. The Flemish clients rush to come and buy woad balls directly from the producers. The competition to buy woad gets so fierce in the region that farmers, for once, can demand to be paid cash and in the best currencies of the time [8] (p. 68).

On the fourth of October 1559, at a time when Etienne Ferrières thinks that with all the cash he has managed to gather together, he will only be able to buy 1,200,000 to 1,300,000 woad balls ("*douze ou treze cent milliers de coquaignes*"); he reckons this will produce 2000 loads of couched woad ("*pour en avoir deux mil charges agranat*"). However, his associates in Lyons soon manage to send him a convoy of mules that travel through the mountains of the Massif Central, loaded with bags of gold and silver coins, so that in the end, the quantity of woad their company has been able to buy that year amounts to the staggering number of 2,400,000 woad balls.

According to Gilles Caster, the historian who first studied the *Registre* extensively, the *charge* mentioned by Etienne Ferrières in connection with his woad transactions weighed 157 kg. Therefore, the 2000 *charges* of couched woad he was expecting to obtain from 1,200,000 to 1,300,000 woad balls would correspond to 314 metric tons. According to this reckoning, the 2,400,000 woad balls that he and his associates eventually managed to buy in 1559 may have produced from 3692 to 4000 *charges*, that is, about 580 to 628 metric tons of couched woad.

It is now possible to compare Ferrières's notes with similar data recently found by Mathieu Harsch in an earlier account book, written from 1362 to 1396, preserved in the Archivio di Stato (State Archive) of Florence. The *Librio in proprio* was a book written by Giovacchino Pinciardi, a woad merchant from Sansepolcro, in Tuscany, who had moved to Florence [4,9]. From Pinciardi's book, it appears that in 1362, the 104,482 woad balls he had imported to Florence from Sansepolcro and the towns of Mercatello and Sant'Angelo in Vado, further east in the hills of the Marches, allowed him to produce 33,653 kg of couched woad (= 33.653 metric tons) [4] (p. 425). Although his business obviously was not of the same magnitude as Ferrières's, the massive quantities of woad balls and couched woad mentioned in these historical documents immediately raise the issues of the huge amounts

of fresh leaves needed and of the area of land that must have been cultivated in woad to obtain them. Interestingly, this is still the first issue that is raised in all discussions concerning the possibility and desirability of a revival of the production of natural dyes at a larger scale today.

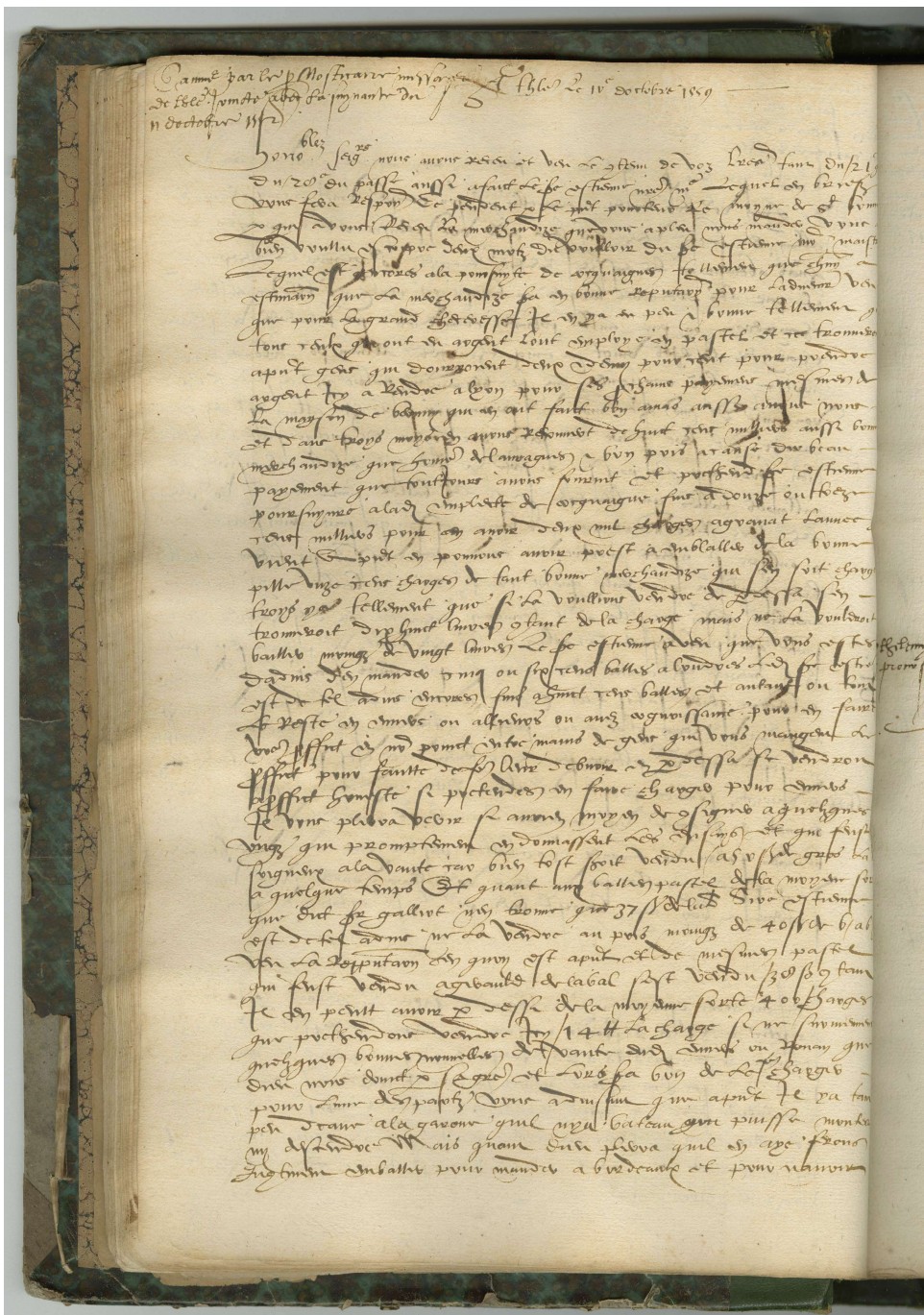

**Figure 1.** A page from Etienne Ferrières's Registre. Credit: Archives municipals de Toulouse—Archives anciennes, ref. HH 61, folio 25 verso.

### 3.2. The Expected Contribution of Experimental Archaeology

The approach of experimental archaeology is a way to obtain plausible answers to the questions raised by the data found in historical documents and to fill the remaining gaps in their descriptions of the production process. The goal is to try and reproduce the whole line of woad production, such as growing woad plants in a certain area and defined conditions; picking and weighing the fresh leaves, whose weight will serve as reference weight (= 100%) in all further calculations; making woad balls and weighing first the fresh and then the dry balls; and lastly, couching the crushed woad balls and weighing the dry weight of dye product obtained.

The aims of this study are to obtain, firstly, quantitative data to establish the weight losses at each successive stage of woad processing and, secondly, propose statistically significant weight proportions of woad at each of these stages, in relation to the weight of fresh leaves produced on a given land area. Finally, to assess the land area required to obtain the production volumes mentioned in Ferrières's *Registre* or Giovacchino Pinciardi's *Librio*. These experiments are the only way to obtain samples of woad at different stages to allow qualitative and quantitative analyses of their indigoid content via HPLC analyses, and to produce enough couched woad to test its coloring power and quality as a source of indigo. This information is necessary to scale up laboratory experiments of the historical vat process to its commercial proportions.

### 3.3. Description of Experiments and Results

Starting in the summer of 1999, the first author already performed a series of experiments in growing woad and making woad balls and couched woad, in preparation for an international exhibition and catalog dedicated to the "Precious Dyes of the Mediterranean: Purple, Kermes, Woad" [10] (pp. 156–159), [11] (p. 370). Additional new experiments were conducted from the spring of 2021 to the autumn of 2022 in order to collect more data and check and complete the first results.

During the two springs, woad was sewn in a row in the first author's garden in the south of France, at an altitude of 650 m above sea level, in granitic, sandy, well-manured soil. Sewing in April allowed for the first harvest of leaves in June and four more pickings to be performed until the end of October. The plants continued to produce beautiful leaves through the winter but these were not collected because it was assumed that the lack of heat and sunlight made them very poor in indigoid precursors. This would still be needed to be studied. The densely sewn plants were thinned down to 6 cm apart in May, as soon as the leaves started forming fairly large rosettes (Figure 2).

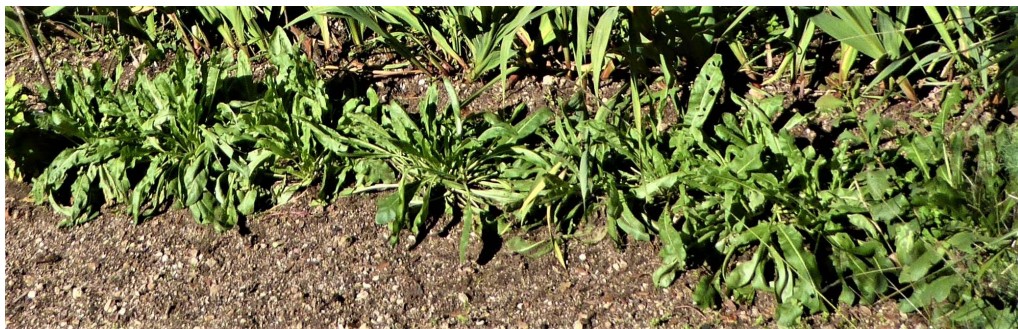

**Figure 2.** Row of woad plants, June 2022 (credit: D. Cardon).

The average yield of fresh leaves per harvest was 1.5 kg per 2.5 m of a row of woad plants. This would correspond to a yield of 6 kg of fresh woad leaves per row that is 10 m long. Assuming plants were cultivated on bigger plots of land, arranged in rows spaced 40 cm apart, this corresponds to a yield of 150 kg per are (= 100 m$^2$). Scaling this up to 1 hectare (10,000 m$^2$), the amount would be 15,000 kg of fresh leaves per harvest (250 rows,

100 m long). The yield from four harvests could, therefore, amount to 600 kg of fresh leaves per are ar per year. Thus, the total yield per year would be 60 tonnes per hectare.

After each picking, the fresh leaves were crushed in a mortar, shaped by hand into big balls, and put to dry in the shade (Figures 3 and 4). By weighing the products of 31 successive pickings, it was possible to calculate that the average weight of the dry woad balls corresponds to 19% of the weight of the fresh leaves from which they were obtained.

In December 2021, all the woad balls produced during the former summer season were crushed with a hand-held wooden mallet and the mass of crumbles was sprinkled with just enough warm water to keep it damp. This was turned over each day for one week (Figures 5–7). From then on, no more water was added, and turning of the mass was only performed every two and then four days for two more weeks during which lumps were formed and gradually became completely dry and hard (Figures 8 and 9). The weight of couched woad obtained amounted to 76.5% of the weight of dry woad balls that were processed.

From these results, it can be calculated that 1 hectare of woad can produce 11,400 kg of dry woad balls and hence, 8721 kg of couched woad per year. This quantity would have allowed the setting of 39 woad and indigo vats in 18th century Languedoc [6] (p. 42).

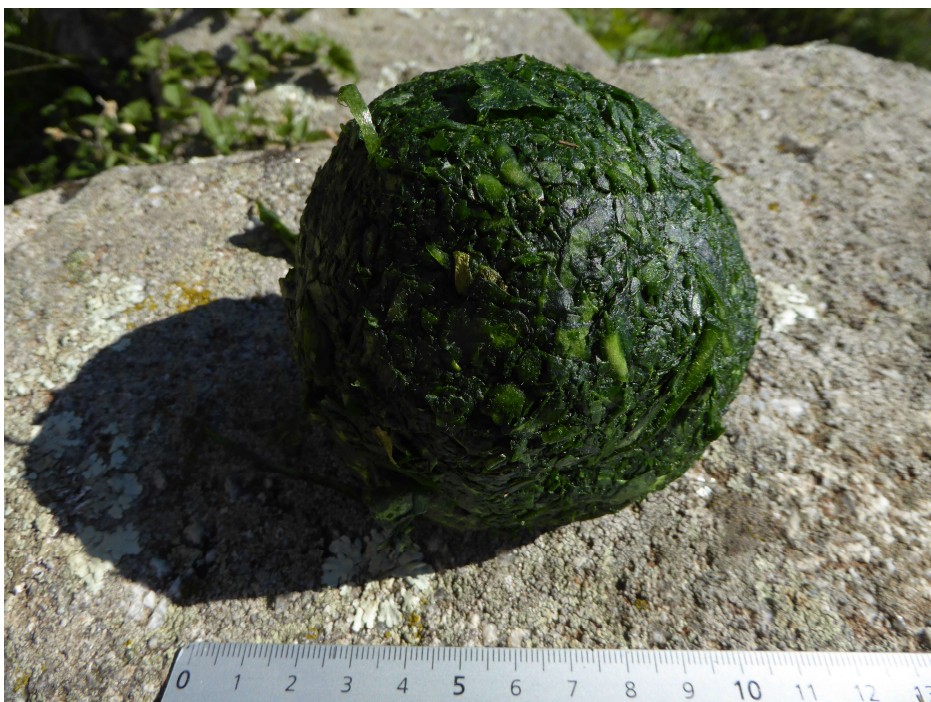

**Figure 3.** Fresh woad ball (credit: D. Cardon).

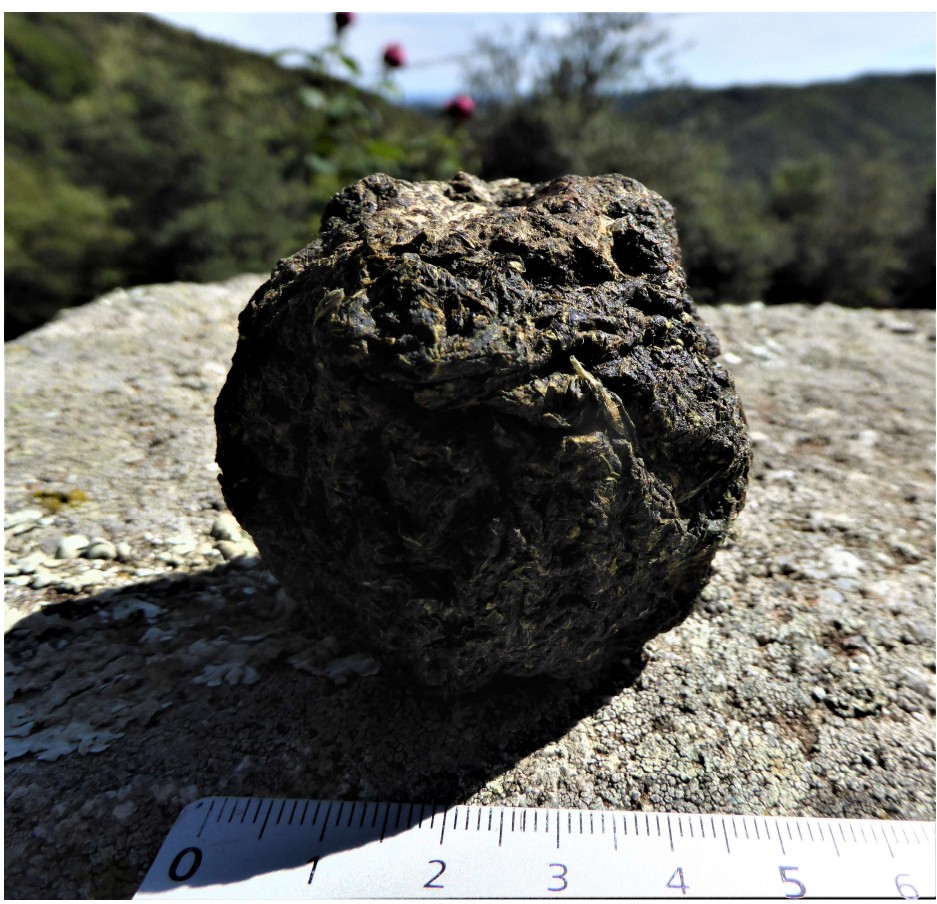

**Figure 4.** Same woad ball as in Figure 3, dry (credit: D. Cardon).

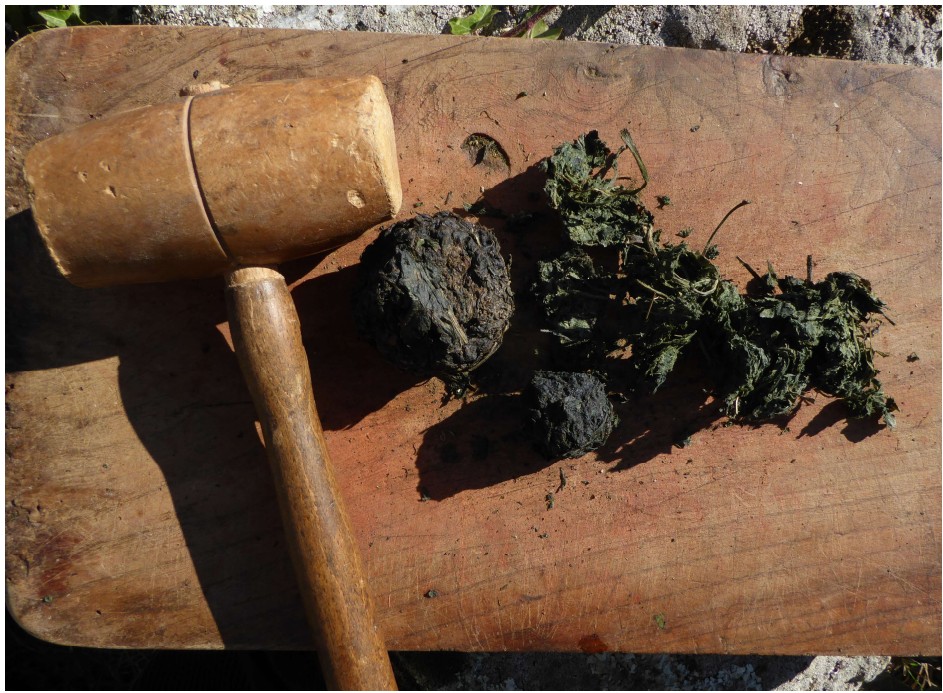

**Figure 5.** Crushing of dry woad balls to make couched woad. Day 1 (credit: D. Cardon).

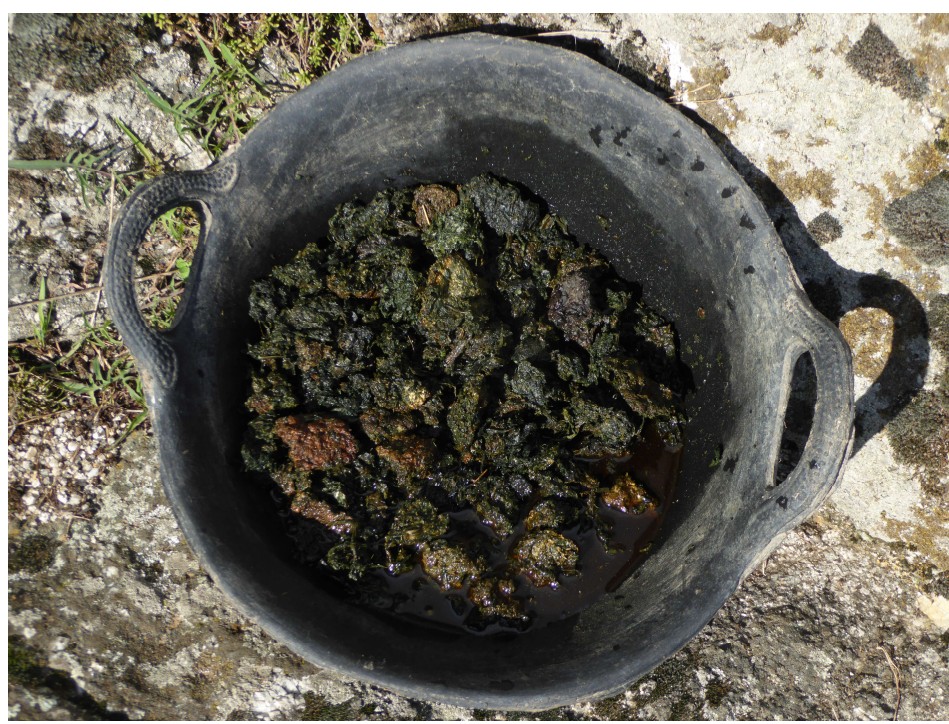

**Figure 6.** Adding enough warm water to thoroughly wet the mass of crushed woad balls. Day 1 (credit: D. Cardon).

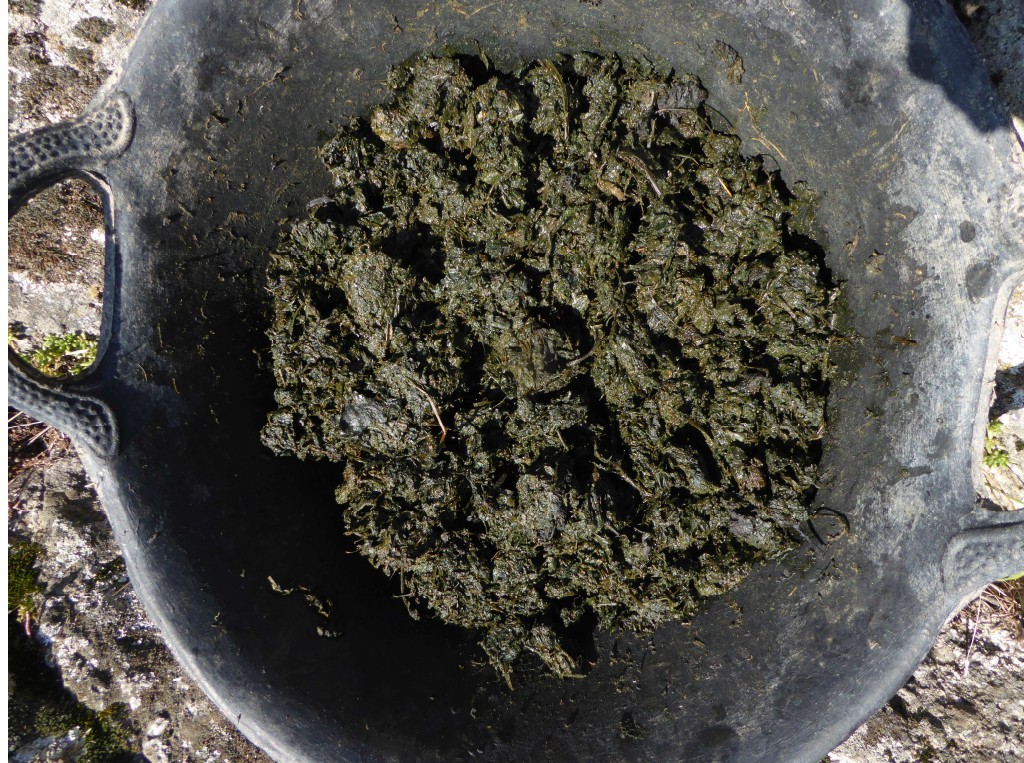

**Figure 7.** The fermenting crushed woad. Day 1 + 5 (credit: D. Cardon).

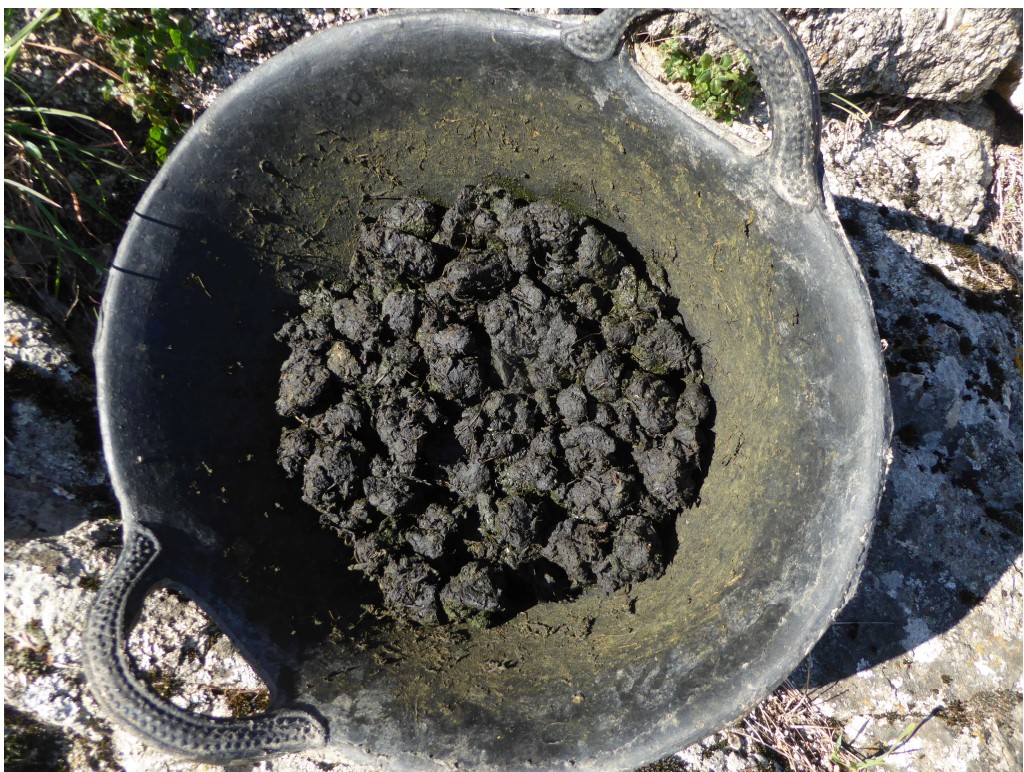

**Figure 8.** Couched woad agglomerating into lumps and drying. Day 1 + 13 (credit: D. Cardon).

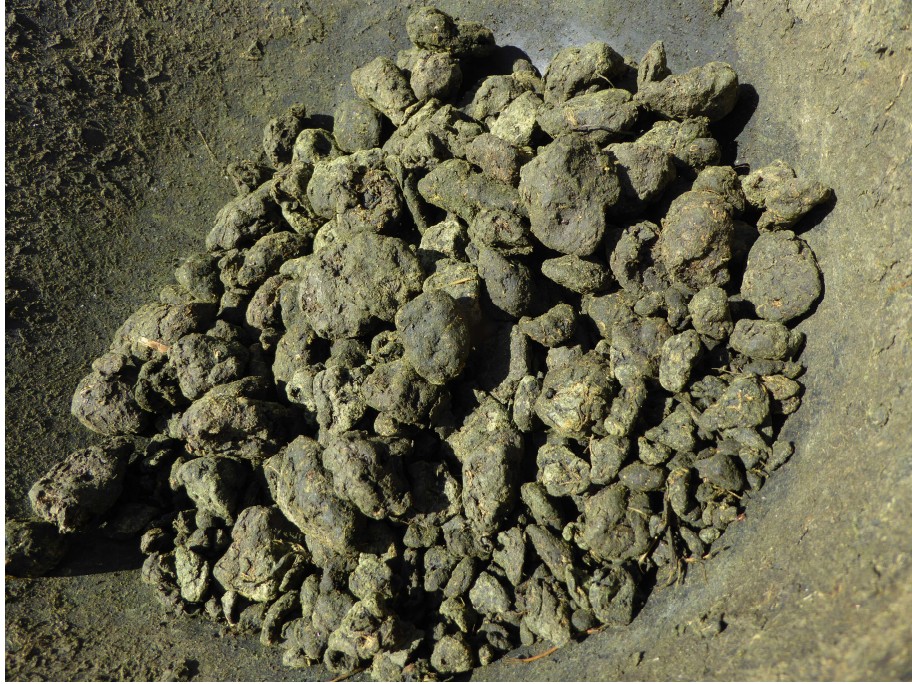

**Figure 9.** Couched woad, final state (credit: D. Cardon).

*3.4. Discussion on Experimental Results in Relation to Etienne Ferrières's Registre*

According to these results, the 580 to 628 metric tons of couched woad collected by Ferrières in 1559 would have required the processing of 758 to 821 metric tons of dry woad balls. To obtain 758 to 821 tons of dry woad balls, 3990 to 4321 tons of fresh leaves would have had to be collected, crushed, and made into woad balls. These amounts, of 3990 to 4321 tons of fresh leaves, could be obtained from 66.5 hectares (= 0.665 km$^2$) to 72 hectares (= 0.72 km$^2$) cultivated in woad, in four harvests per summer.

The somewhat surprisingly high yield of couched woad per area of woad cultivation, based on these experiments, is in agreement with the results of recent historical research and contributes to explaining some newly collected data from historical primary sources by the French historians Marie-Claude Marandet and Mathieu Harsch. The latter found that in 1362–1363, in his *Librio in proprio*, Giovacchino Pinciardi recorded that he could obtain 36,122 pounds (in Florence) of couched woad from 49,300 pounds of dry woad balls, which corresponds to a proportion of 73.26%—quite close, interestingly, to the 76.5% obtained in the course of the first author's experiments [4] (pp. 238–239). Marandet, studying local notaries' books, observed that in the region of Lauragais, famous for its woad production, in the first half of the 15th century woad was cultivated around many villages, mostly on comparatively small plots of land of sizes commonly between 3000 and 6000 m$^2$, with one mention of a bigger parcel of 2.4 hectares. These could be worked by the members of one family who would also bring the leaves to the nearest mill—if they did not own one—shape the woad balls and put them to dry in sheds. Growing woad in this way, almost like a garden crop, proved highly profitable, as highlighted by a contemporary chronicler, Guillaume de Catel, who wrote that "the profit gained from woad in Lauragois is so great that it often happens that in a fertile year, the product of a woad field is worth as much as, or even more than, the price of the field in which it has been sewn". This assertion has been confirmed by Marandet's calculations, based on data mentioning the selling prices of land and of woad balls that she could find in local contemporary sources [3] (pp. 52, 57).

## 4. Re-Assessing the Efficiency of the Woad and Indigo Vat in Light of Biochemical Research

*4.1. Antoine Janot's and Paul Gout's Memoirs on Dyeing*

Antoine Janot, in 1744, and Paul Gout, in 1763, wrote *Memoirs* on piece dyeing of wool broadcloth with fast dyes, illustrated with dozens of cloth samples dyed in colors corresponding to the processes described (Figures 10 and 11). Both manuscripts have been recently published as critical editions and the descriptions of their dye recipes have been translated into English [12–15]. Their accounts allow woad production to be followed down the line to the use of couched woad in the woad and indigo vat. Since the 17th century, when the addition of imported indigo pigment to the woad vat was allowed in European countries, this came to be the process generally used to obtain the various degrees of fast indigo blues on wool and woolen textiles.

Antoine Janot was a master dyer, and owner of a dye workshop in Saint-Chinian, a small town but an important center of wool broadcloth production, north of Béziers, in Languedoc. Paul Gout was the manager of two Royal Manufactures of fine broadcloth in succession, first in Saint-Chinian from 1754 to the end of 1756, and then in Bize, not far to the south of Saint-Chinian, from 1756 to the first years of the French Revolution. In both places, he also acted as a master dyer, supervising all dyeing operations. In Antoine Janot's dye workshop, an average of 3000 pieces of cloth, produced by the local clothiers of Saint-Chinian and its surroundings, were dyed per year. Paul Gout's Royal Manufacture of Bize was producing between 600 and 800 pieces of cloth annually. As in all the textile centers of Languedoc at the time, mostly Londrin Second broadcloth, the best-selling quality in the Levant, was produced and all pieces were exported to different parts of the Ottoman Empire. Each cloth piece had to measure from 18 to 20 m in length, be at least 1.40 m in width, and weigh no less than 10.35 kg. Nearly half of them had to be dyed in the woad and indigo vat into various shades of blue. Depending on the Oriental customers' orders,

some pieces were kept blue but a majority of blue-dyed pieces were further mordanted and top-dyed in red or yellow dye baths to produce beautiful ranges of purples and greens. Blue grounds of different degrees were also used to create numerous shades of greys and browns. In addition, dark blue grounds were key prerequisites to obtaining fast black dyes. All the above helps us understand at what considerable scale couched woad was routinely used in the dye workshops of these broadcloth production centers.

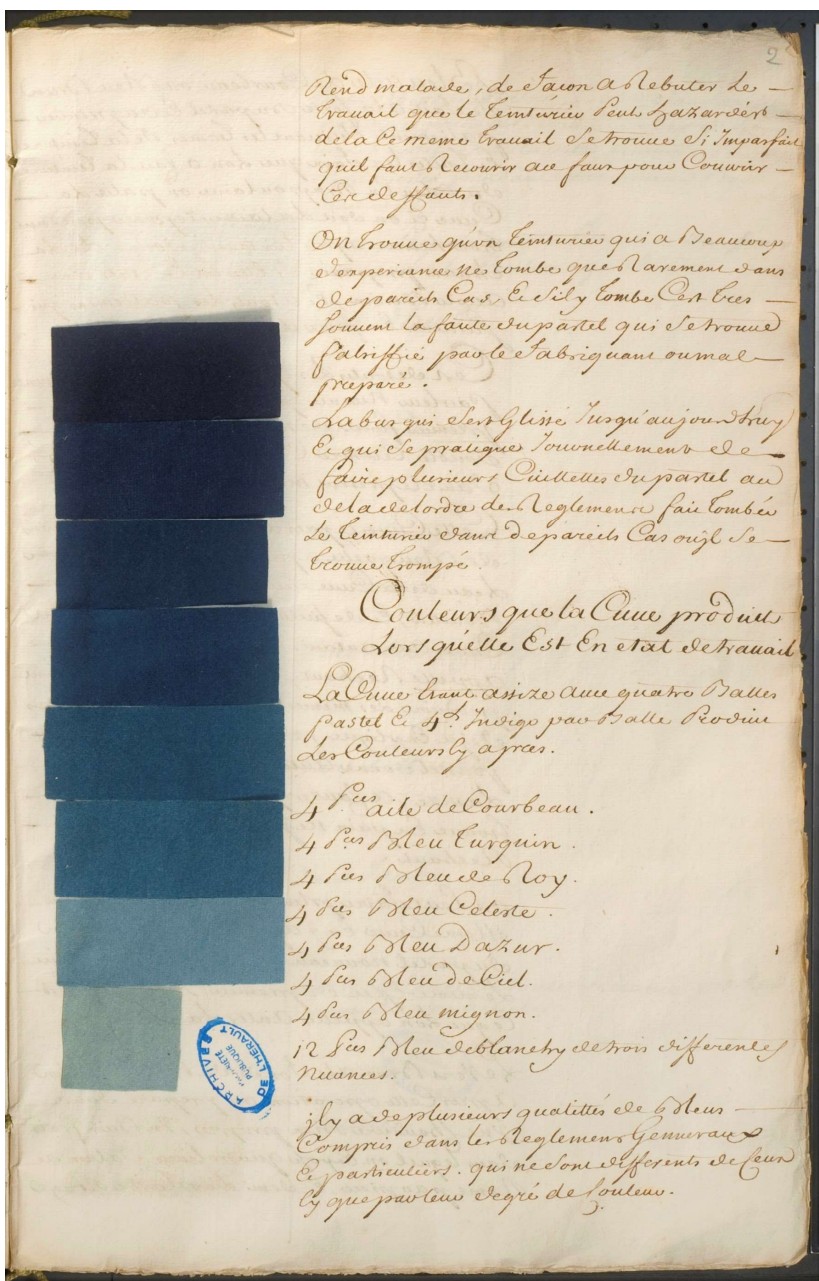

**Figure 10.** Antoine Janot's scale of blues from the woad and indigo vat. Memoir on dyeing, 1744. Archives départementales de l'Hérault, ref. C/5569, folio 2 recto (credit: D. Cardon/P.-N. Granier).

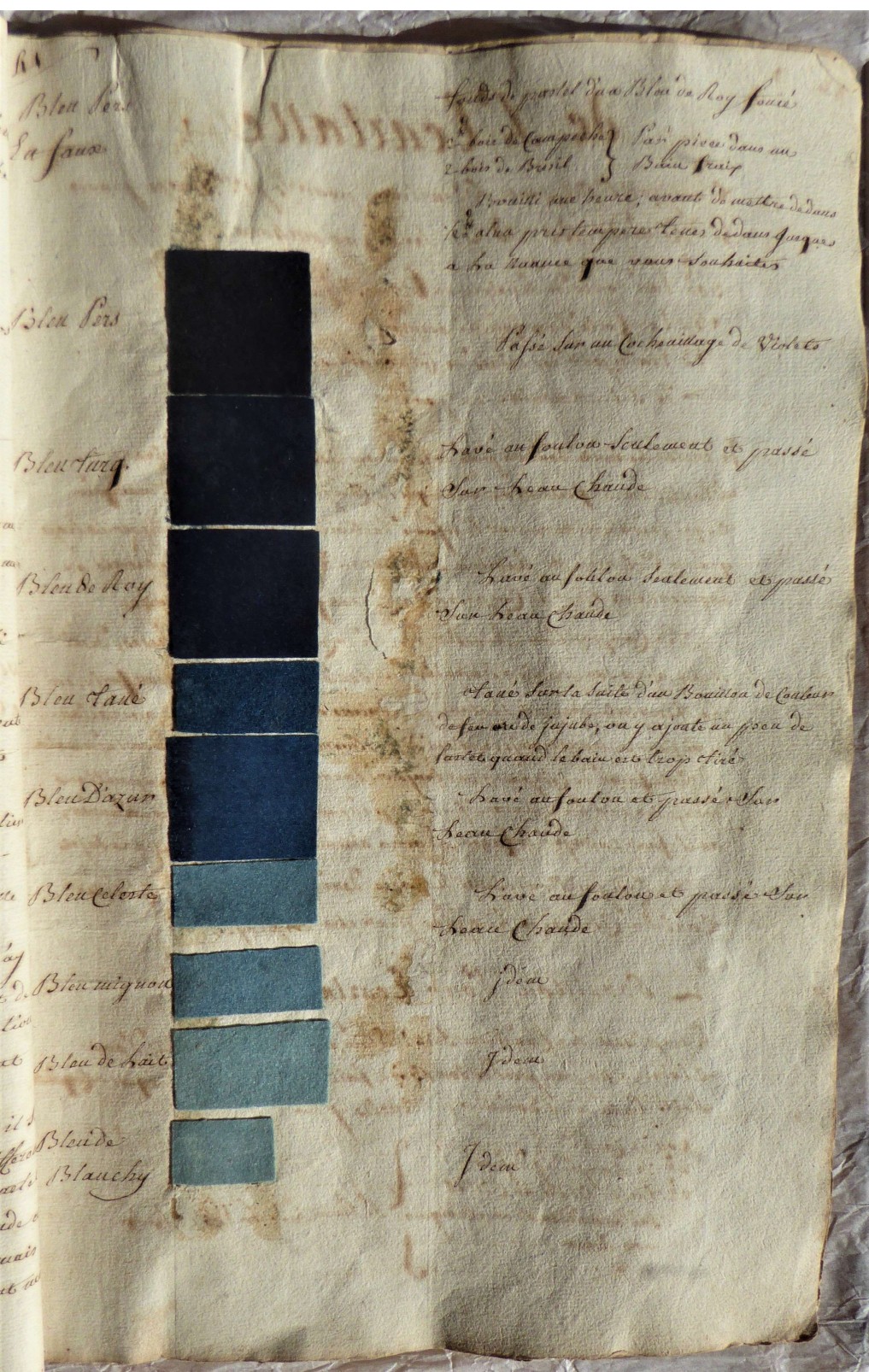

**Figure 11.** Paul Gout's scale of blues from the woad and indigo vat. Memoirs on dyeing, 1763. Priv. Coll., p. 41 (credit: D. Cardon).

In his description of the setting of a vat, Gout mentions that three bales (about 223.56 kg) of couched woad are used [6] (p. 42). Indigo pigment, imported from the French West Indies, is only added later, when "the vat is ready and that it bears some blue [ . . . ]

imperceptible veins can then be seen: in such a state, the vat can receive such quantity of indigo as one judges appropriate to put into it" [15] (p. 52). Antoine Janot gives the more precise proportion of 4 pounds of indigo powder to be added, per each bale of woad (= 2.2% of the weight of couched woad used). Both Gout and Janot are clearly aware of the double function of couched woad in the process; firstly, as the main fermenting agent giving the dye bath its reducing power, essential in every vat-dyeing process (Figure 12), and secondly, as a source of blue colorant.

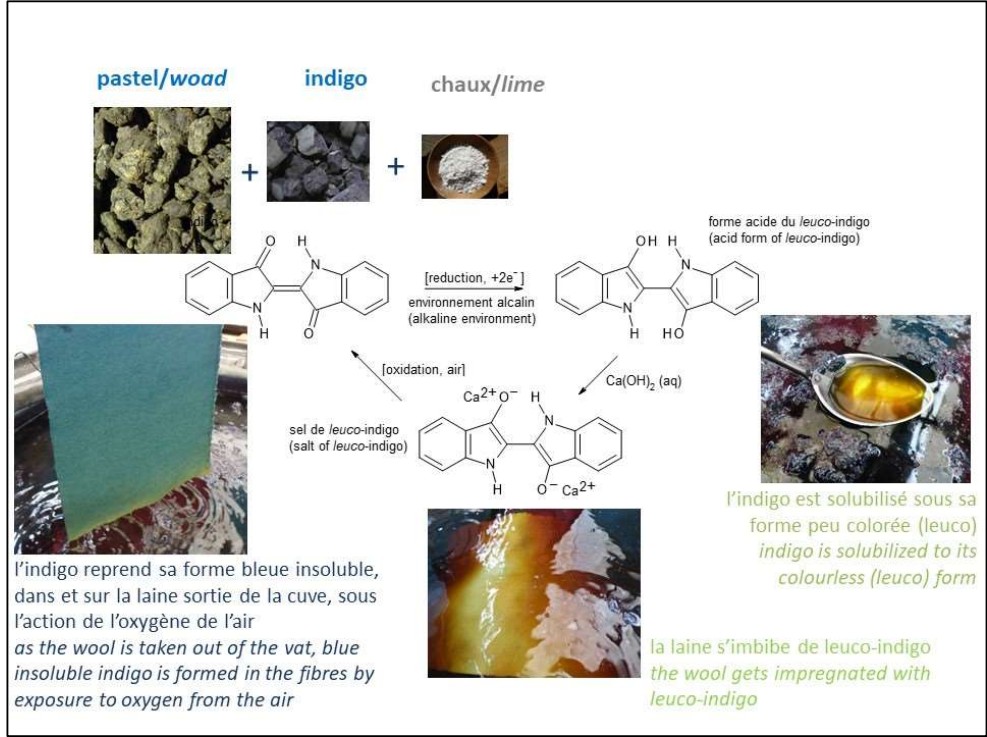

**Figure 12.** Chemical reaction of indigo in the woad and indigo vat of 18th c. dyers. Indigo is solubilized and reduced in the vat, imbibes the textile, and is regenerated in it by the oxygen from the air (credits: photo D. Cardon, D. Santandreu; diagram D. Cardon, C. Higgitt).

Consequently, our research and experiments focused on better understanding the scientific basis of the two distinct contributions of couched woad: the satisfactory evolution and the coloring efficiency of the vat. Hisako Sumi included the couched woad produced by Dominique Cardon in her research, pursued in collaboration with bacteriologists in Japan. They studied the bacteriological environment that gives a woad and indigo vat its power to reduce the insoluble blue indigo pigment to its soluble form, to impregnate the textile fibers. Zvi Koren's chemical analyses explored the contents and proportions of indigoids in the woad leaves from the moment they are collected to their reduction into dry-couched woad.

### 4.2. Bacteriological Environment and Its Reducing Role in the Fermentation Liquor of a Woad and Indigo Vat

Hisako Sumi's experiments with woad and indigo vats were parts of a wider research, aimed at comparing four types of indigo-fermented vats, their reducing power, and the colors obtained [16–18].

In the first experiment at a reduced scale, a 20-L vat was set (Figure 13), using 1275.5 g of couched woad and 41.4 g of crushed woad balls (total weight 1316.9 g) produced by D. Cardon, as described above. Then, 3 L of wood ash lye (measured pH 10.87), heated to 35 °C, were added. After 2.5 h, the pH decreased to 9.59 and 5 g of slaked lime was added to obtain the necessary higher degree of alkalinity mentioned in reports on previous

experiments to reproduce the medieval woad vat [19]. The next day, the pH decreased to 8.94, and 15 L of wood ash lye and 35 g of slaked lime were added, as well as 90 g of wheat bran boiled in water for 15 min. After stirring, the pH of the liquor was checked again and had gone up to 10.33. On the morning of the third day, the alkalinity of the liquid was measured again and found to be at pH 9.29; a purple film was forming on the surface of the vat (Figure 14). Slaked lime was added to keep the pH close to 10 and the temperature of the bath was maintained at about 30 °C. One hour later, the first dyeing test produced a good medium blue, but after a rather long oxidation time (Figure 15). The reason for this was thought to be a combination of the high reduction performance of the microflora of the woad fermentation liquid and its low indigo content. In this experiment, natural indigo pigment from India was added 8 days after setting. "As per the dyer's judgement", following Gout's approach, the first addition consisted of 65 g of indigo cakes reduced to powder (5% in relation to the weight of couched woad used). A purple layer and fine bubbles started forming on the liquid surface (Figure 16). An additional quantity of 5% indigo powder was added later. From then on, dyeing could be performed repeatedly for several months (Figure 17). The temperature of the vat was kept as constant as possible, as well as the optimum alkalinity, maintained by successive additions of slaked lime.

In later experiments, the volume of the vat was first increased to 50 L and then to 100 L, re-using the same indigo fermentation liquor and adding new woad balls produced in Hokkaido, wood ash lye, wheat bran, and indigo pigment. Details of all experiments have already been published elsewhere [16,17,20]. These experiments have shown that woad and indigo vat-related microorganisms are particularly active at pH 9.8–10.3, with temperatures of the vat liquor kept between 25 and 30 °C, which is lower (less energy-consuming) than in other organic vats.

The bacteriological environments of the vat liquors from these experimental reconstitutions of the woad and indigo vat were studied in Hokkaido, Japan, by a team of bacteriologists. This study was part of more general research into the bacterial flora of various indigo fermentation vats set by H. Sumi with diverse indigo plant sources, processed in different forms. An indigo-reducing bacterium, *Clostridium isatidis*, had already been identified by a scientific team in England, in the fluid of a couched woad vat prepared following a medieval Italian recipe published by D. Cardon [19,21]. The woad and indigo vats set by Hisako Sumi were found to contain a particularly rich bacterial flora. A complete list of the species identified has been published elsewhere [22] (p. 283, Figure 2). They include two different indigo-reducing species: *Alkalibacterium indicireducens*, and, in the fermentation liquor of Sumi's second experiment, two distinct strains of a new species of indigo-reducing bacteria were isolated and described as *Fundicoccus fermenti* sp. nov. [23]. The presence of these indigo-reducing bacteria in the liquors of woad and indigo vats contribute to explaining why these vats could become ready for dyeing in such a short time and retain their reducing power for such a long time, provided the right environmental conditions for the proliferation of the useful bacteria were maintained.

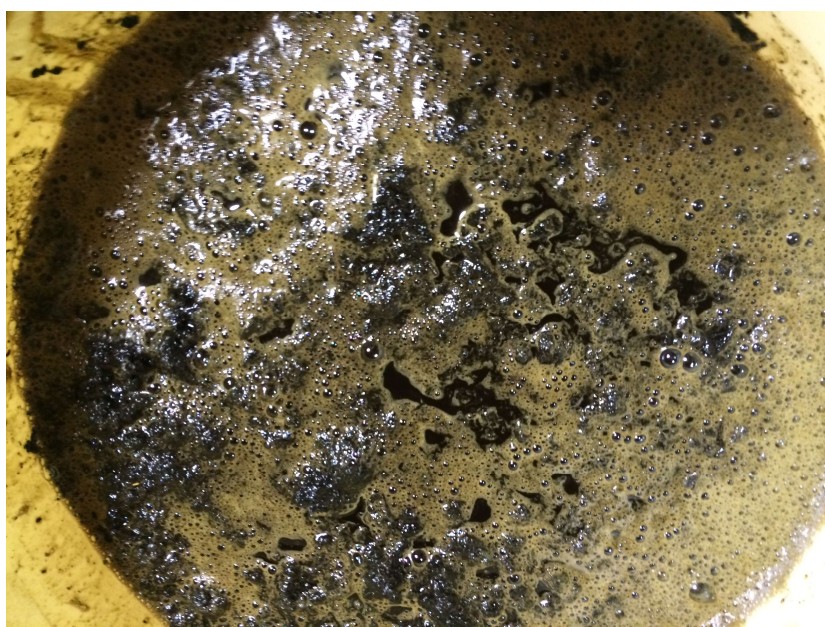

**Figure 13.** Setting of a woad and indigo vat in Hisako Sumi's Studio in Otaru, Hokkaido, Japan, the first day (photo H. Sumi).

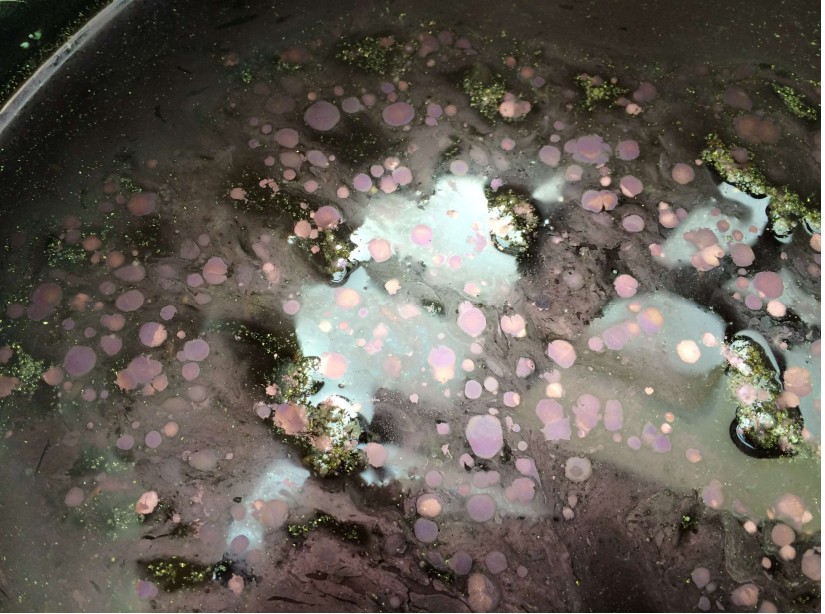

**Figure 14.** The third day, morning. Patches of reduced indigo from woad reoxidizing at the surface of the vat (photo H. Sumi).

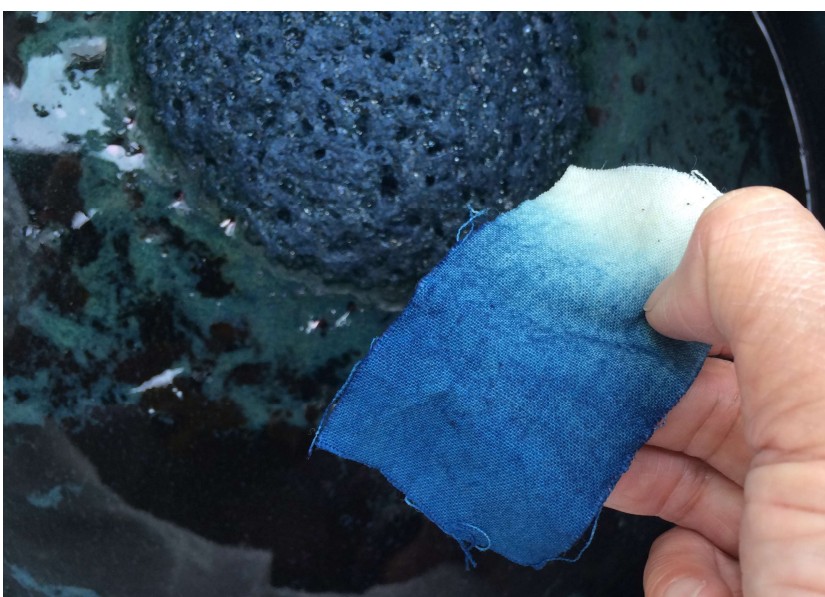

**Figure 15.** The third day, later. Testing of the coloring power of the vat with a sample of cotton cloth (photo H. Sumi).

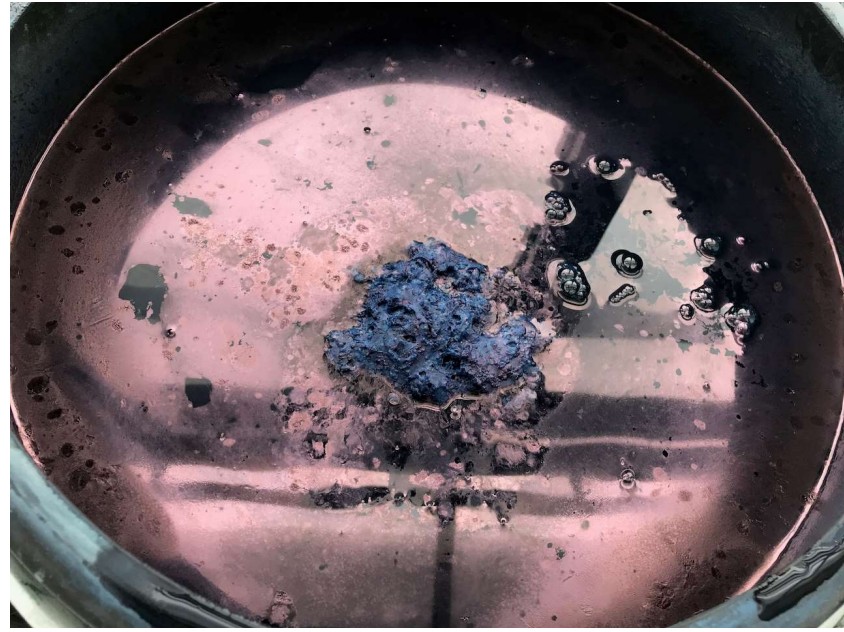

**Figure 16.** Appearance of the top of the woad and indigo vat after additions of indigo pigment (photo H. Sumi).

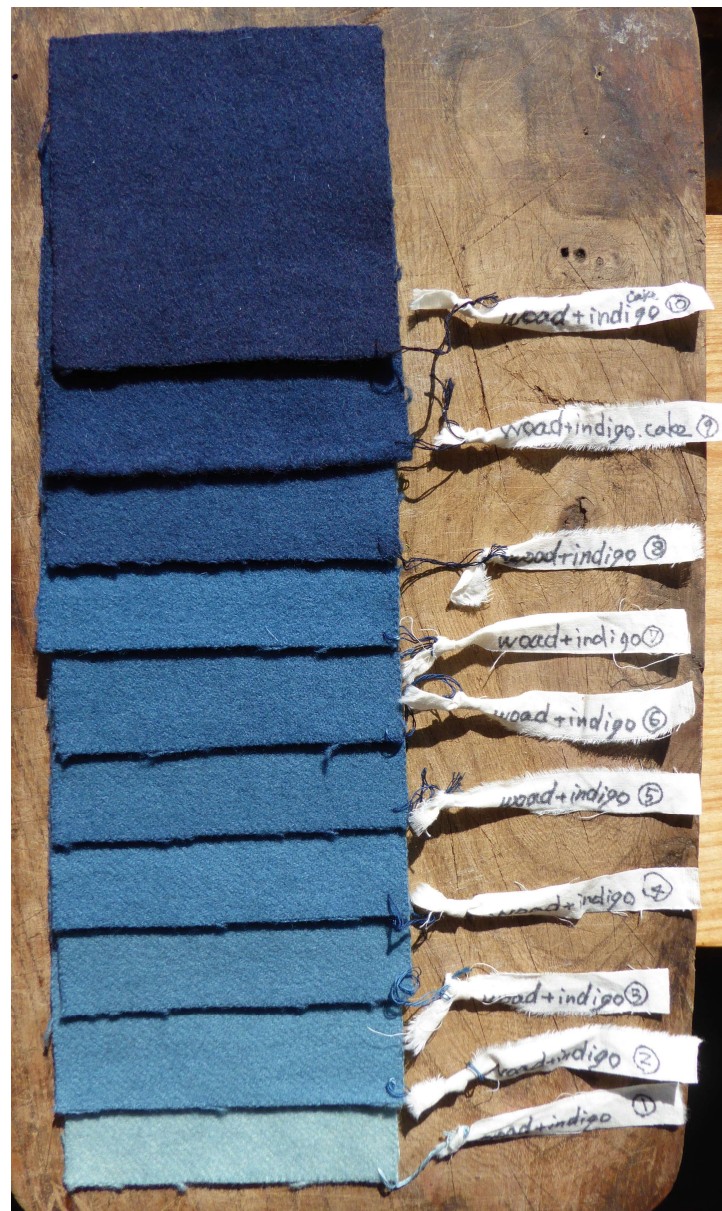

**Figure 17.** Hisako Sumi's scale of blues from the woad and indigo vat (photo D. Cardon/H. Sumi).

*4.3. Woad as a Source of Indigo Blues: An assessment via HPLC Analyses*

The following results were obtained as part of a more general study [24] aimed at quantifying the dye components in indigo-producing plants. This report focuses on the leaves of *Isatis tinctoria* L. (woad), and for comparison purposes, on *Persicaria tinctoria* (Aiton) H.Gross whose leaves, like those of woad, were traditionally processed into a compost (Japanese *sukumo*). Woad was cultivated in Dominique Cardon's garden in the mountains of Cévennes, south of France. The couched leaves of *P. tinctoria* were provided by artist dyer Hisako Sumi from Hokkaido, Japan.

Prior to the dye extraction, each dried plant sample was pulverized in a mortar and pestle in order to produce a relatively uniform powdered mass for the analyses, and fresh leaf samples were cut into smaller pieces. Each sample analyzed consisted of an exactly weighed quantity (usually a few milligrams). The dyes from these samples were extracted via repeated additions of measured volumes (usually a few milliliters) of dimethyl sulfoxide (DMSO) at 100 °C for 5 min at a time until no visible color in the last extracted solution was visible. The multiple extractions from each sample were combined and highly concen-

trated extracted solutions were diluted when necessary. Prior to the HPLC analysis, the dye solution was filtered in a centrifuge tube assembly consisting of a 0.2-µ nylon filter.

The ambient-temperature reverse-phase Waters chromatographic system used consisted of a 600E Controller pump and a 996 PDA detector, each controlled by the Millenniu m-32 software. The stationary phase was composed of a Waters 3.0 × 150 mm C-18 Symmetry column with 5 mm and 100 Å particle and pore diameters, respectively. The ternary mobile phase eluents passed through a 20 µL sample loop and consisted of the following: methanol, water, and phosphoric acid ($H_3PO_4$, 5% aqueous). The HPLC elution method for the separation of indigoids and related dyes consisted of a constant flow rate of 0.8 mL/min. In addition, a constant 10% volume of acid was used, and thus water decreased inversely with the following increasing % volumes of methanol:

0–3 min: 30–60%; 3–14 min: 60%; 14–21 min: 60–90%; and 21–40 min: 90%.

### 5. Results of the HPLC Analyses

A typical HPLC chromatogram showing the three main dye components investigated in this study—indigo (IND), indirubin (INR), and isatin (IS)—is shown in Figure 18. The peaks eluting from 4–12 min are non-colorants extracted from the leaves and are thus not discussed in the paper, which focuses on the three main dyes investigated in this study.

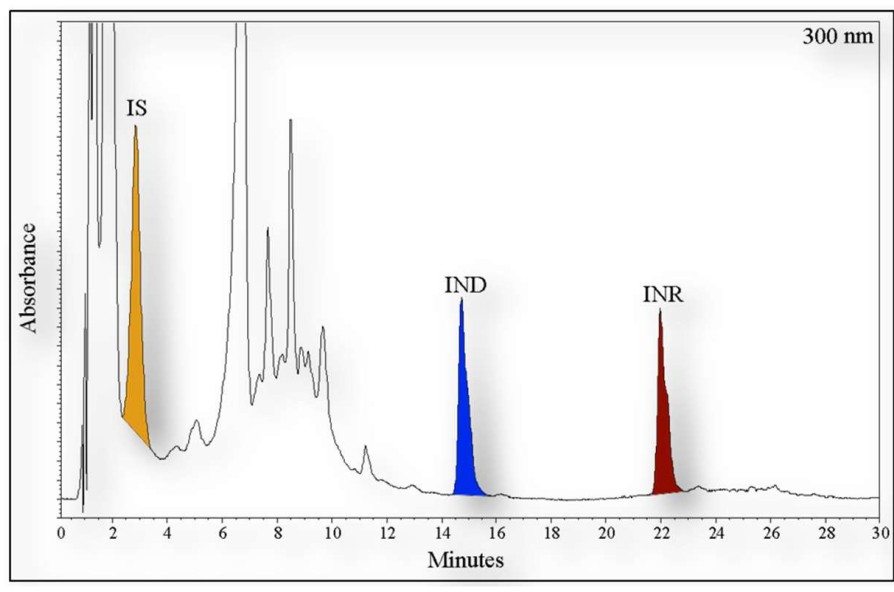

**Figure 18.** Typical HPLC chromatogram (at 300 nm) displaying the separation of the three dyes: isatin (IS), indigo (IND), and indirubin (INR).

The UV/visible spectra and molecular structures of these dyes are given in Figure 19.

The results of the HPLC analyses are based on the PDA-derived integrated peak areas (in PDA units) in the chromatograms calculated at a specific wavelength. The wavelength chosen for the quantification of the presence of IS, IND, and INR was the near-UV value of 300 nm because at this wavelength all three dyes have appreciable light absorptions. These dyes also have appreciable absorptions at wavelengths below 300 nm, for example at about 240 nm; however, at these low UV values, a significant number of co-eluting colorless components in the leaves have relatively high absorptions, which may mask that of the dyes.

In order to prevent the inclusion of any co-eluting components in the peak area calculations of the dyes, the contour diagrams were first checked to determine whether there are any co-eluting components. Secondly, the peak areas were calculated in the chromatogram displayed at their respective visible $\lambda_{max}$ values and then converted to the standard UV wavelength value, which was the method previously used [25]. Thus, for indirubin, the

wavelength of 540 nm was chosen for the calculation of the integrated peak area because it is the only colorant extracted from the leaves that has a maximum absorption at that wavelength, and any possible co-eluting non-colorants at the same or similar retention time are invisible at that wavelength. Similarly, for indigo, the peak area was first calculated at 615 nm. The peak area values at 300 nm for both dyes were then calculated based on the absorbance ratios for each dye at their visible $\lambda_{max}$ relative to their absorbance at 300 nm. These relative absorbance ratios are obtained from the UV–Vis absorption spectrum of each dye. For isatin, its peak area was evaluated with the chromatogram displayed at 241 nm, as at this value, its absorbance is extremely high, as can be seen in its UV–Vis absorption spectrum, and then was scaled down to the standard 300 nm as with the other dyes.

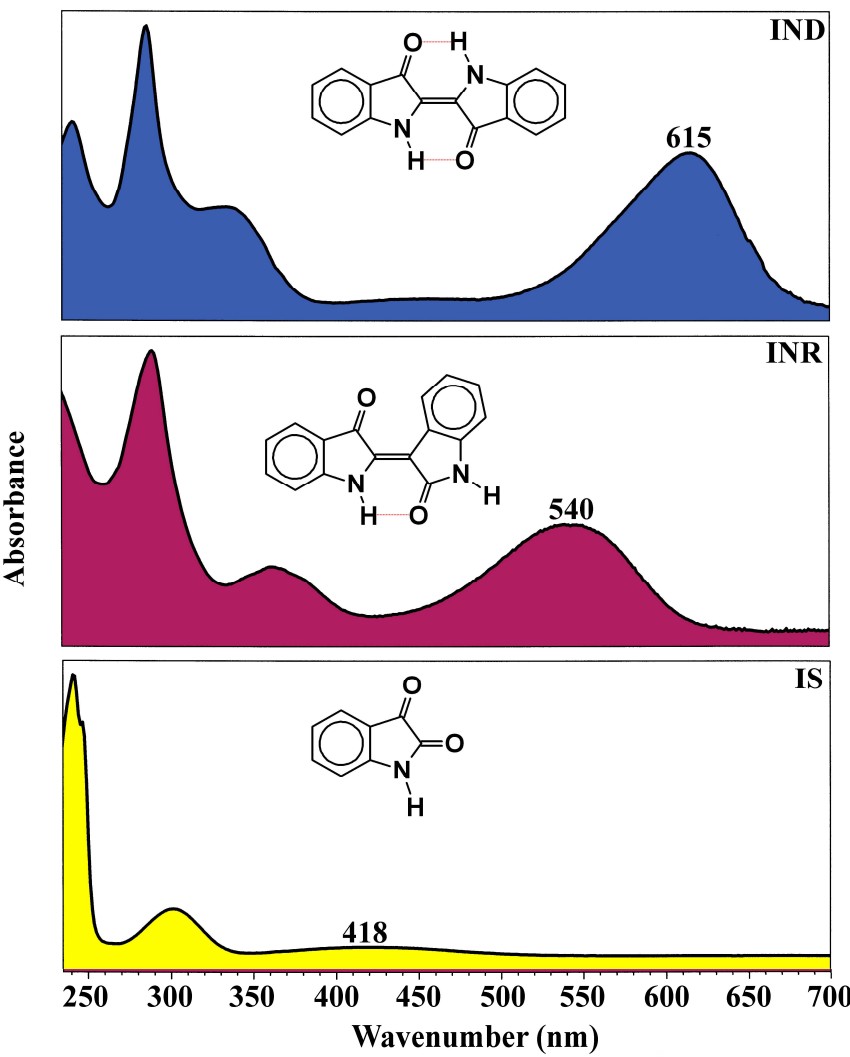

**Figure 19.** UV/Vis spectra and molecular structures of the three dissolved dyes as obtained by the PDA detector, showing the prominently visible wavelengths with maximum absorption, $\lambda_{max}$.

These peak area values were normalized with regard to the constitution of the injected dye solution by dividing the area values by the composition. The composition of the injected dye solution is represented by the mass of the leaf sample divided by the total volume of DMSO used (including all dilutions). The resulting scores given for each dye in the figures below are thus these normalized peak areas (measured at 300 nm) and scaled

to represent convenient values typically greater than 1. Thus, the compositional value for each dye *i* can be expressed as:

$$\text{composition of dye } i = 10^{-11} \times \text{Area}_{i,300}/(g \text{ of material}/\mu L),$$

where the $10^{-11}$ factor is to down-scale the values to represent more practical numbers.

It should be noted that these peak areas are not percentage values, nor actual compositions of the dyes in the leaves, but are related to them and can, nevertheless, be used as semi-quantitative comparative properties indicative of relative compositions, as discussed below.

### 5.1. Fresh and Aged Wood Leaves

The following triplicate analyses of each sample assess the indigoid contents of different woad materials: fresh leaves, dry leaves, dry woad balls, and couched woad.

Figure 20 shows the quantitative HPLC results of woad leaves stored in paper and allowed to age up to 67 days after cutting. Comparing the dark and green areas of relatively fresh leaves, 3 days after cutting and storing in paper (labeled as 3 d and 3 g, respectively) shows that the indigo content in the dark area was about 10 times that of the green area, which was also accompanied by nearly 25 times more isatin. An appreciable amount of indirubin was also detected in the dark area and was absent in the green part of the leaf. Allowing the leaves in paper to age for 17 days after cutting showed a respective increase in all the dyes. Indigo increased by about 25%, a nearly threefold increase in indirubin content, and there was 70% more isatin. After 67 days of aging, the indigo content in the dark dry leaves essentially did not change with only a slight increase in indirubin, while the quantity of isatin decreased to about a third of its previous level.

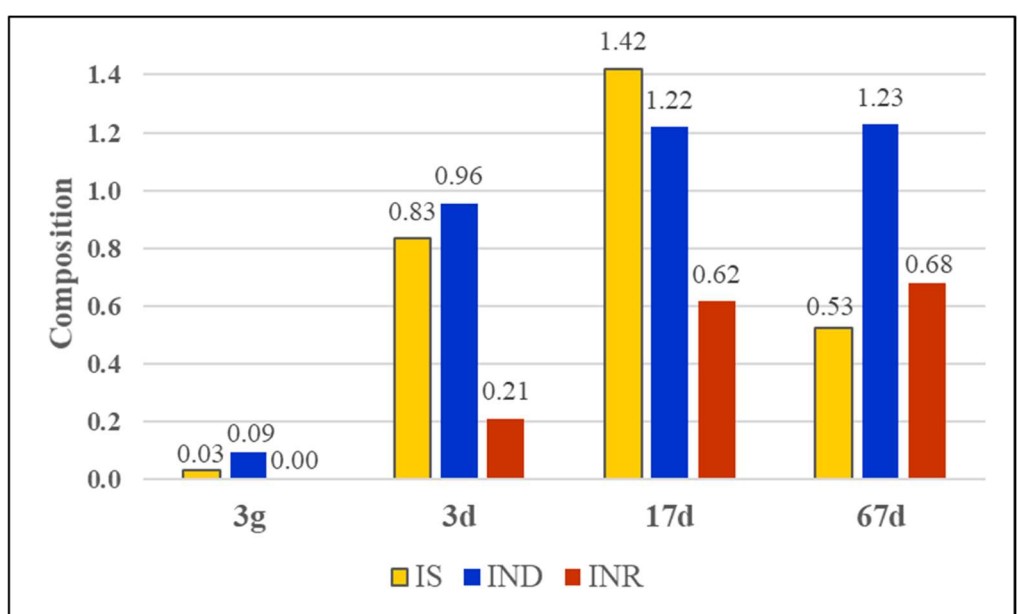

**Figure 20.** Indigo content in woad leaves stored in paper 3 days after cutting in the green part of the leaf (3 g) and in the dark part (3 d), and the dark part of the leaf after 17 days (17 d) and 67 days after cutting (67 d). (See text for explanation of "composition").

The effect of the way woad leaves are stored also has an effect on the dye compositions in the leaves. This can be seen from the first two sets of Figure 21, which compare the green parts of woad leaves stored in paper vs. those stored in plastic, both 3 days after cutting. Although, in both cases, the quantity of indigo in the green leaves is, as expected, very low, the indigo content in the paper-stored leaf is almost twice that of storing the leaf in plastic. This phenomenon may be due to the availability of oxygen in the air in the case

of paper as opposed to that of hermetically sealed plastic, which effectively does not allow atmospheric oxygen to enter the plastic bag. Thus, there is more oxidation of the precursors in the leaves to indigo in the paper case than in the plastic one.

Processing of woad can increase the formation of indigo in the leaves, as shown in Figure 21, which shows the comparison for woad leaves that were processed into a ball vs. woad that has been couched. Figure 19 shows that couching has a markedly strong effect on the indigo content in the processed woad by more than 6-fold. Couching is very efficient in concentrating indigo. Although isatin and indirubin contents also show significant increases, their overall quantities are very small.

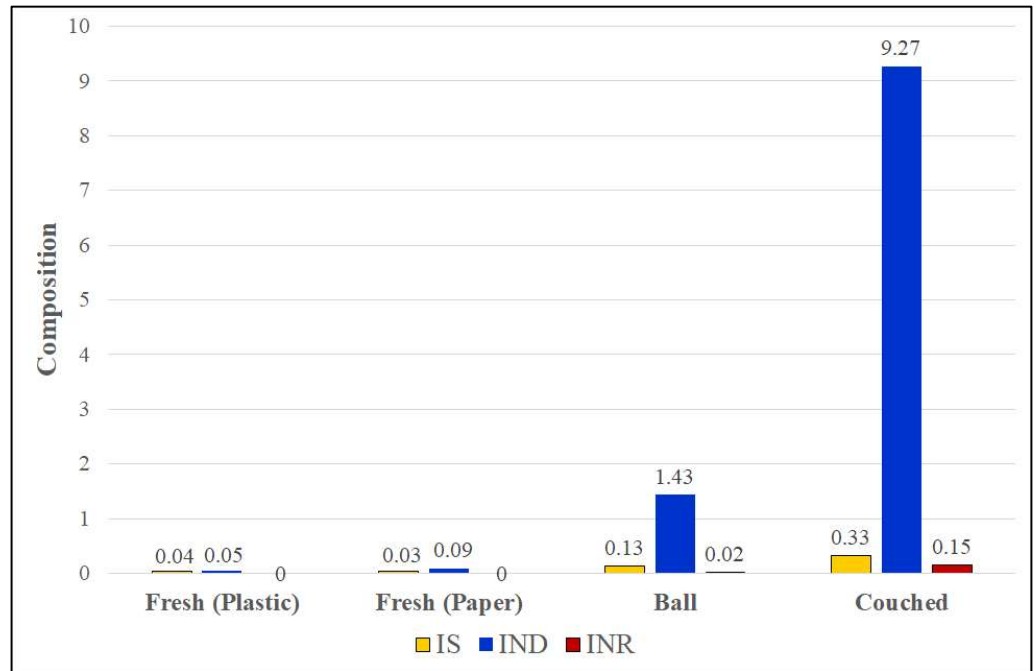

**Figure 21.** Indigo content in green woad leaves stored in paper compared with those stored in plastic 3 days after cutting (first two sets in the graph) and dye content in woad material formed into a ball vs. couched (last two sets in the graph).

### 5.2. Comparison of Couched Woad and Sukumo

A comparison between the indigo concentrations in couched woad with another processed plant material, *Persicaria tinctoria* compost (*sukumo*), is shown in Figure 22.

Although couching of woad increases the indigo content in that plant material, as shown above, the couched woad is a less concentrated source of indigo than, for example, *sukumo*. Thus, assuming that after the crushing and couching of these plant leaves all the precursors in the leaves have undergone enzymatic hydrolysis (as a result of these actions) and subsequent oxidation to form indigo, in the plant material used in this study, there is about 2.5 times more indigo in the crushed *P. tinctoria* than in couched woad.

On the other hand, H. Sumi's experiments with both the traditional Japanese *sukumo* vat and the woad and indigo vat have shown that in a 100 L vat, 6 kg of *sukumo* is needed, while 2 kg of couched woad with the later addition of 640 g of indigo powder are enough to obtain both a good reducing power and a stronger coloring power. An additional advantage is the lesser volume of plant colorant mass at the bottom of the vat, leaving more space for the textiles to dye evenly.

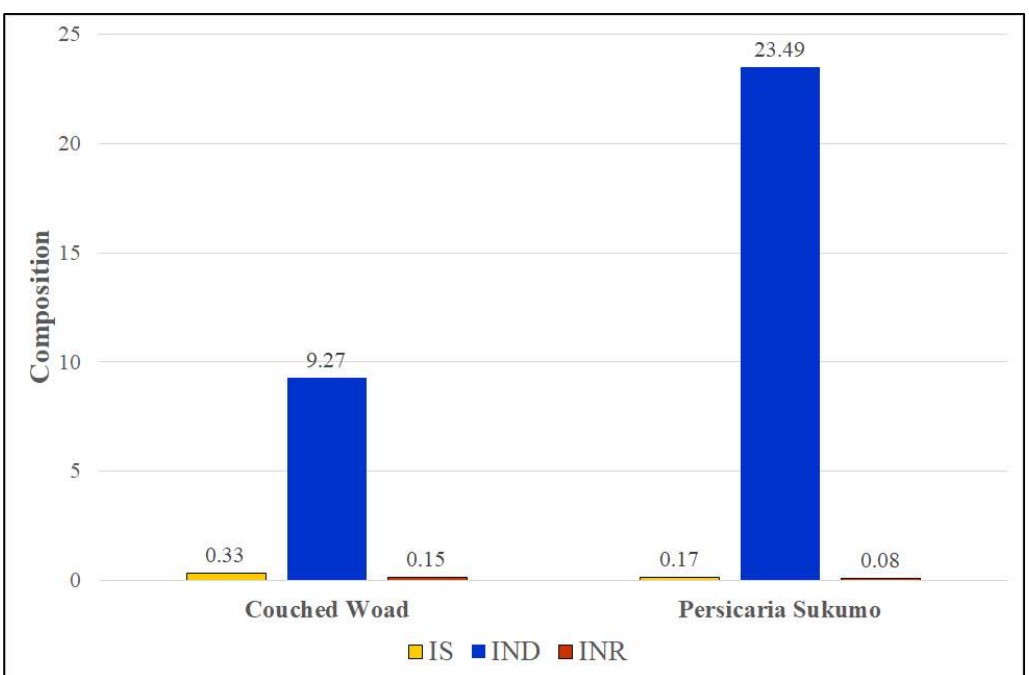

**Figure 22.** Indigo content in couched woad vs. in *Persicaria tinctoria* compost (*sukumo*).

## 6. Conclusions

Growing woad plants as a garden or small plot crop was found to give higher yields of fresh leaves per year than recorded from large-scale mechanized farming, as performed from the latter part of the 19th c. to the recent experiments at reviving woad cultivation in order to produce woad indigo pigment.

The processing of woad leaves into couched woad, as opposed to the maceration process leading to the production of woad indigo pigment, appears an efficient way of obtaining a good yield of indigoid colorants from woad. Moreover, couched woad has been found to significantly contribute to the reducing power of the woad and indigo vats. The woad and indigo vat process was found to be a cost-efficient and environmentally friendly process only requiring warm water and the addition of wheat bran, wood ash lye, and slaked lime to the two sources of indigo.

In the conclusions drawn from the HPLC analyses, it is important to note that in this study, the quantities of all the precursors in the leaves were not investigated. The analytical HPLC study focused on the spontaneous formation of three detectable dyes—indigo, indirubin, and isatin—in the leaves of these plants. The analytical results lead to the following conclusions:

1. Spontaneous indigo formation begins even in the green part of the leaf as soon as the leaf is cut off from the plant.
2. With time, further oxidation significantly increases the indigo content in the dry parts of the cut leaves with an eventual decrease in isatin as more of it is converted to indirubin.
3. Couching of woad significantly increases the indigo content in the dried leaves.
4. However, couched woad does not yield the maximum amount of indigo, which is the case with other plant species.

Our experiments and their corresponding scientific analyses validate the efficiency of the whole technological line of woad production and use as practiced in Europe since the Middle Ages.

**Author Contributions:** Conceptualization, D.C.; methodology, D.C., H.S. and Z.C.K.; software, D.C. and Z.C.K.; validation, D.C., Z.C.K. and H.S.; formal analysis, D.C. and Z.C.K.; investigation, D.C.,

Z.C.K. and H.S.; resources, D.C., H.S. and Z.C.K.; data curation, D.C., H.S. and Z.C.K.; writing—original draft preparation, D.C., Z.C.K. and H.S.; writing—review and editing, D.C. and Z.C.K.; visualization, D.C. and Z.C.K.; supervision, D.C.; project administration, D.C.; funding acquisition, none. All authors have read and agreed to the published version of the manuscript.

**Funding:** This research received no external funding.

**Data Availability Statement:** All data are provided in the manuscript.

**Acknowledgments:** Z.C.K. is extremely grateful to the other co-authors for supplying him with the various plant leaves and is also most appreciative of the support given to his research on natural dyes by the Sidney and Mildred Edelstein Foundation.

**Conflicts of Interest:** The authors declare no conflict of interest.

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
