# Peer review of "Woaded Blue: A Colorful Approach to the Dialectic between Written Historical Sources, Experimental Archaeology, Chromatographic Analyses, and Biochemical Research"

_heritage, doi:10.3390/heritage6010037_

Round 1
Reviewer 1 Report
This is a very interesting paper to demystify pertinent questions about traditional methods of European woad for modern production and dyeing. The novel aspects are quantification of the agricultural yield of couched woad from 14th- 18th century dyeing documents in France and Italy, a bacterial study under controlled conditions to explain the reduction rate and sustainability of a traditional dyeing vat, and chemical analysis as an indicative measure of the colorant content in couched woad from European and Japanese plants. The results provide new practical insight regarding land use and preparation of couched woad as a traditional dyestuff.
Some changes and clarifications are required, I hope the following comments and suggestions are useful.
Image credits needed for Figs 1-12
Lines 27 Mention the two plant species studied (Isatis and Persicaria). Clarify that a traditional Japanese woad vat sukumo was created with woad for the study.
Line 56 Clarify that the woad was couched in France and used in a traditional Japanese dye vat.
Lines 62 and 97-98 Remove brackets
Line 68 Give the date and origin of the historical documents used: 14th C to 16th C France and Italy
Line 99 Add the date/period of indigo import
Line 92 suggest ‘the same imagery is used in sukumo, the Japanese word for couching.’
Lines 121-126 appears to be a translation from the manuscript so should be in quotations
Line replace ‘appeared as’ with ‘is’
Line 167 suggest ‘and, secondly. propose’
Line 171 delete ‘not only’
Line 172 replace ‘but also’ with ‘and’
Lines 173-174 suggest ‘… as a source of indigo. This information is necessary to scale up laboratory experiments of the historical process to its commercial proportions.’
Line 180-181 Replace with ‘Additional new experiments were conducted.’ Suggest ‘During the springs of 2021 and 2022,…’
Lines 185-186 change present tense to past
Lines 198-202 Recommend calculations in hectares first and then the 1/1000th scale-down to the less familiar ‘are’. Suggest ‘Assuming plants were cultivated on bigger plots of land, arranged in rows spaced 40 cm apart, this corresponds to a yield of 150 kg per 100 m2. Scaling this up to 1 hectare (10000 m2), the amount would be 15,000 kg of fresh leaves per harvest (250 rows, 100 m long). The yield from four harvests could therefore amount to 600 kg of fresh leaves per are per year, or 60,000 kg. Thus the total yield per year would be 6 tonnes per hectare.’
Line 215 say that it was crushed with a hand-held wooden mallet
Line 216 Suggest deleting ‘and’ and replace with ‘This was turned over every day’. Was warm water added every day too?
Line 260-261 Remove brackets
Line 279 suggest ‘Their accounts allow woad production to be followed down the line… ’
Line 334 add that this was the environment of Cardon’s woad balls
Line 335 delete ‘, able’
Line 369 Hisako Sumi’s
Line 410 Check line spacing
Line 419 It would seem difficult to pulverise fresh samples. How was this achieved?
Line 444-446 Fig 18 Please comment here on the peaks eluting 4-12 mins that absorb at 300 nm.
Line 457 The sloped baseline of IS and tailing of the IND and INR peaks indicate co-elution of other 300nm-active compounds. Please comment on their peak purity for the area calculations.
Line 491 Please give the number of repeat extractions
Line 504 Fig 20. Decimal point rather than ‘,’ for numbers.
Line 519 -522 ‘The dried balls… ferment and dried’ is repetition of information and can be deleted
Line 556-557 What is the yield and processing by maceration for modern late-19th C farming of woad? It is useful to mention it in the introduction. Would it be indigo at this time (see line 60)?
Reviewer 2 Report
This article is an excellent example of a multidisciplinary approach to tackling not only a historical question but also to proposing a sustainable solutions to modern dye industry. A combination of historical research, experiments, microbiology and analytical chemistry provide a novel approach and new answers to seemingly well-research question of woad production in post-Medieval Europe. The quantitative data resulting from experimental growing is particularly welcome from an economic history perspective.
The article is very well structured and clearly written and I recommend it for publication.
One tine comment to line 109: "1559 to 1561 AD" - AD traditionally precedes the date, so "A(nno) D(omini) 1559-1561".
Reviewer 3 Report
The paper is very interesting and well written, the topic presents good references with the actuality in terms of sustainability and it can provide good references to the actual research both in dye chemistry and to sustanable processes.
Two minor comments are provided in the attached pdf.
